# COVID-19 Peritraumatic Distress as a Function of Age and Gender in a Spanish Sample

**DOI:** 10.3390/ijerph18105253

**Published:** 2021-05-14

**Authors:** María Pilar Jiménez, Jennifer A. Rieker, José Manuel Reales, Soledad Ballesteros

**Affiliations:** 1Departamento de Psicología Básica II, Universidad Nacional de Educación a Distancia, C/Juan del Rosal 10, 28040 Madrid, Spain; mpjimenez@psi.uned.es (M.P.J.); jrieker@psi.uned.es (J.A.R.); 2Departamento de Metodología para las Ciencias Sociales, Universidad Nacional de Educación a Distancia, C/Juan del Rosal 10, 28040 Madrid, Spain; jmreales@psi.uned.es

**Keywords:** COVID-19 Peritraumatic Distress Index, cluster analysis, exploratory factor analysis (EFA), confirmatory factor analysis (CFA), age differences, psychological distress, psychological impact

## Abstract

The sudden outbreak of the COVID-19 pandemic has profoundly altered the daily lives of the population with dramatic effects caused not only by the health risks of the coronavirus, but also by its psychological and social impact in large sectors of the worldwide population. The present study adapted the COVID-19 Peritraumatic Distress Index (CPDI) to the Spanish population, and 1094 Spanish adults (mean age 52.55 years, 241 males) completed the Spanish version in a cross-sectional online survey. To analyze the factorial structure and reliability of the CPDI, we performed an exploratory factor analysis (EFA) followed by a confirmatory factor analysis (CFA) on the Spanish sample. The effects of gender and age on the degree of distress were analyzed using the factorial scores of the CPDI as the dependent variables. Results showed that, after rotation, the first factor (*Stress*
*symptoms*) accounted for 35% of the total variance and the second factor (*COVID-19 information*) for 15%. Around 25% (*n* = 279) of the participants experienced mild to moderate distress symptoms, 16% (*n* = 179) severe distress, and about 58% (*n* = 636) showed no distress symptoms. Women experienced more distress than men (p<0.01), and distress decreased with age (p<0.01). We conclude that the CPDI seems a promising screening tool for the rapid detection of potential peritraumatic stress caused by the COVID-19 pandemic.

## 1. Introduction

The outbreak of a new coronavirus has caused a “tsunami” that has profoundly altered the daily lives of large sectors of the world population, with dramatic effects caused not only by the health risks of COVID-19, but also by its psychological and social impacts. The World Health Organization [1] renamed it COVID-19 and on 11 March 2020 declared a state of pandemic. During the first wave of the pandemic, Spain was particularly affected by the spread of the virus. The first case was reported on 28 February and the curve of new cases increased exponentially. To reduce the risk of viral transmission at the national level, the Spanish Government (BOE, 2020) [2] declared a state of emergency from 14 March to 26 April 2020. During this period, citizens were allowed to go out only for the purchase of essential items (e.g., food, medicine), attend health centers, go to work (only for jobs considered essential, such as food suppliers), return to usual residence, care for dependents such as young children or disabled family members, and other force majeure events. In April, the curve of people infected with COVID-19 began to flatten, with 7940 reported new cases per day at the beginning of April, decreasing to 1631 by the end of the month [3]. Measures to reduce the spread of COVID-19 led to a disruption of normal daily life, social interactions, employment, and leisure activities.

Based on research on the psychological impact of previous outbreaks of coronavirus infections, such as SARS and MERS, the occurrence of psychological distress and symptoms of mental illness have been outlined in recent articles [4,5]. A recent review [6] on the psychological impact of lockdown highlighted its negative, potentially long-lasting impact on psychological health. These negative effects can be amplified by stressors occurring both during and after lockdown (e.g., duration, lack of information, financial loss).

At the beginning of the COVID-19 outbreak, most of the published articles were from China [7]. Using a survey procedure, Wang et al. [8] revealed that 53.8% of respondents reported a psychological impact of the outbreak, 16.5% reported moderate to severe depressive symptoms, 28.8% reported moderate to severe anxiety symptoms, and 8.1% reported moderate to severe stress levels. In a nationwide survey (*n* = 52,730), Qiu et al. [9] reported that 35% of the respondents experienced mild to severe COVID-19 peritraumatic distress. The first systematic review and meta-analysis, conducted with 17 articles, revealed a high prevalence of stress (29.6%), anxiety (31.9%), and depression (33.7%) in the general population [10]. More recent meta-analyses reported a prevalence rate of 25% of anxiety [11] and depression [12]. The prevalence of depression increased during the pandemic as much as sevenfold [12], stressing the impact of COVID-19 on mental health.

The first study conducted in Spain during the first stage of the COVID-19 outbreak explored the extent to which different variables (i.e., age, degree of concern, significant changes in daily life, environmental conditions, or leisure activities during lockdown) were associated with the psychological impact (anxiety, depression, and stress) caused by the pandemic. In regard to stress and depression, Spaniards showed moderate to severe levels (22% and 30%, respectively). Women and young adults were those who experienced the greatest psychological impact. The lower the household size, the better the mental health [13]. In another study [14] conducted in the north of Spain, more than a quarter of the participants reported symptoms of depression (27.5%), anxiety (26.9%), and stress (26.5%). Furthermore, the study revealed that stress, anxiety, and depression levels were higher when measured two to three weeks after the beginning of lockdown. In relation to gender, men showed higher levels of depression than women. Moreover, young adults (18–30 years) and middle-aged adults (31–59 years) showed higher levels of stress, anxiety, and depression than older adults (60–82 years). In relation to the variables that could have a predictive character of psychological stress in a Spanish population, Gómez-Salgado et al. [15] reported that being female, working outside the home, the perception of poor health, living in close contact with an infected person, and the number of symptoms are the variables that have most impact on psychological distress. Imperative preventive measures implemented by Governments to control the spread of the virus also become a source of stress, leading to anxiety and extreme distress.

Most studies focusing on the mental health outcomes of the COVID-19 pandemic used self-report questionnaires that assessed anxiety, depression, and posttraumatic stress symptoms [16]. In many cases these were well-established and validated instruments that do not explicitly relate to the pandemic situation. This makes it difficult to distinguish between already existing mental health conditions and those produced by the pandemic. Other studies used tools developed “ad hoc”, which limit the generalizability of the results. A third alternative is the use of pandemic-specific scales. These scales allow a rapid detection of COVID-related stress symptoms, facilitating more efficient resource allocation of public mental health support by prioritizing assistance to the most affected population groups. Some of these questionnaires are the Fear of COVID-19 Scale (FCV-19S) [17], the Coronavirus Anxiety Scale (CAS, Lee, 2020) [18], the COVID-19 Phobia Scale (C19 P-S [19], and the COVID-19 Peritraumatic Distress Index (CPDI [9], see [20] for a review. Within these tools, the CPDI probably captures the COVID-related psychological distress most comprehensively, as it incorporates symptoms of a broad array of mental health syndromes and pathologies [20].

The CPDI has been used in several countries, keeping in mind that the predictors of distress during the COVID-19 pandemic may vary across different cultures. Ramasubramanian et al. [21] explored the psychological impact in Indian citizens and reported that 77.2% of the respondents had low or no distress, 20.2% had mild to moderate stress, and only 2.7% had severe stress. In a study conducted in Iran [22], the mean score of CPDI was higher than the results reported in China; 47% and 14.1% of the Iranian adults experienced mild to moderate and severe psychological distress, compared to 29.3% and 5.1% respectively in China. In Germany, Liu and Heinz [23] reported an average CPDI score (Mean = 21.9, SD = 12.6) 1.8% lower than that of the above-mentioned China-based sample (Mean = 23.7, SD =15.4). In the German sample, 24% of respondents reported psychological distress (20.6% with mild stress and 3.6% severe stress), substantially lower rates than in the Chinese sample where 34.4% of the respondents reported distress (29.3% mild and 5.1% severe). According to the authors, this difference might reflect a comparatively higher COVID-19 testing rate in Germany, which may confer a sense of security. In Italy, Constantiny and Mazzotti [24] assessed the prevalence of peritraumatic distress using the CPDI to compare their results with those of Qiu et al. [9]. The CPDI score obtained in the Italian male subsample was slightly lower than that of the Chinese (China, mean = 21.41, SD 15.93 vs. Italy, mean = 18.61, SD = 12.20), but the mean scores of the female subsamples were very similar. Overall, it seems that being female constitutes an important risk factor for developing COVID-related stress symptoms. A recent Italian large-scale study with 20,158 participants found that higher CPDI scores were significantly associated with gender, being two-fold in women in comparison to men [25]. Similar results were found in Chinese [9], Brazilian [26], and Bangladeshi [27] samples. Furthermore, it seems that the clinical distress scores remain constant over time. In a longitudinal study, Megalakaki and colleagues [28] compared the CPDI responses of French participants from the initial phase of the pandemic, when societies were overwhelmed by the sudden outbreak, with those provided several months later, when strategies were established to deal with the situation. As expected, once societies regained a certain control over the situation, the overall CPDI scores declined at follow-up. However, after accounting for the effect of the baseline levels, the CPDI scores still predicted 43% to 47% of the prevalence of posttraumatic stress, depression, and anxiety, confirming its utility for the detection of potentially clinical mental health issues.

Regarding the psychometric properties of the CPDI, several studies from different countries provided the Cronbach’s alpha for internal consistency and reliability [21,22,23,24,25,26,27,28,29,30]. Megalakaki et al. [28] reported the Cronbach’s alpha in a French sample for pretest and posttest measures, together with the stability effect. On the other hand, Costantini and Mazzotti [24] reported the construct validity data for the CPDI in an Italian sample, which was analyzed by means of the correlations between the CPDI scores and three dimensions (Intrusion, Avoidance and Hyperarousal) of the revised Impact of Event Scale [31]. The development of the CPDI was based on diagnostic guidelines for specific phobias and stress disorders and includes items relating to anxiety, depression, specific phobias, cognitive change, avoidance and compulsive behavior, physical symptoms, and loss of social functioning [9]. Given the variety of symptoms assessed in the questionnaire, and despite the widespread use of the CPDI scale, it is surprising that the factorial structure of the CPDI remains largely unexplored. To our knowledge, only one study has conducted a confirmatory factor analysis on the CPDI to validate the tool in an Italian population [32].

Given that the CPDI could constitute a promising tool for a rapid detection of mental health issues in the population, the present work had three main objectives: (1)To investigate the factorial structure of the CPDI via exploratory (EFA) and confirmatory factor analysis (CFA) and validate the questionnaire in a Spanish sample;(2)To quantify the prevalence and severity of COVID-related psychological distress in the Spanish population; and(3)To analyze age and gender differences in relation with the factorial structure of the questionnaire.

## 2. Materials and Methods

### 2.1. Participants

Participants were recruited through postings on social media and snowball sampling. The lockdown measures imposed during the pandemic did not interfere with the data collection process. Inclusion criteria were to reside in Spanish territory and to be more than 18 years old. Exclusion criteria was not to have completed the CPDI. Of the 1463 participants who began the questionnaire, 1094 completed it, yielding a response rate of 74.8%. Thus, the final sample was composed of 1094 Spanish adults (241 males, M_age_ = 52.55, SD = 14.19, range = 18–83 years). Table 1 provides a description of the socio-demographic variables of the sample.

### 2.2. Instruments

To assess psychological distress, we used the Spanish translation of the COVID-19 Peritraumatic Distress Index, CPDI [9]. This instrument is a self-report questionnaire with 24 items that assess COVID-19 peritraumatic distress symptoms. The questionnaire was developed to evaluate COVID-19 distress in China. The authors provided us with the English version, which was translated into Spanish using the forward-backward translation procedure. This method consists in re-translating the translated text back into the source language. No significant inconsistencies were found between the back-translation and the original document. The questionnaire examines the frequency of anxiety, depression, specific phobias, cognitive change, avoidance, compulsive behavior, physical symptoms, and loss of social functioning in recent weeks. Participants rate each item on a 5-point Likert scale ranging from 0 (never) to 4 (most of the time). Scores are summed (total score range 0–100) with higher scores indicating higher COVID-19 peritraumatic distress. Chinese normative data reveal the following ranges for the total score: 28–51 mild to moderate distress and ≥52 severe distress. The CPDI showed satisfactory reliability and content validity.

### 2.3. Procedure

We used a cross-sectional survey design to assess the participants’ psychological response to the COVID-19 pandemic using an anonymous online questionnaire. The online survey was prepared on the Qualtrics Software [33] and responses were collected from 8 May to 25 June 2020. The survey was distributed via an anonymous link to the survey URL and anyone who clicked on the link was able to take the survey. A progress bar indicated the percentage that had been completed and respondents could save and continue the survey for one week. The link could only be used once per participant. The survey data was recorded automatically by the Qualtrics platform. All the participants read a description of the study protocol before responding to the survey and were informed that they could discontinue their participation in the study at any point.

### 2.4. Ethical Considerations

The study was conducted in accordance with the Declaration of Helsinki. The Ethical Committee Board of Research of the Universidad Nacional de Educación a Distancia (UNED) approved the study. The approval document was signed by the President of the Committee and Vice-Rector of Research of the University. Participation was entirely voluntary. All subjects gave their informed consent for inclusion before they participated in the study and gave their explicit informed consent for the confidential use and processing of the data, in accordance with the current laws regarding the protection of personal data [34]. Data were stored anonymously with an assigned number so that it was not possible to identify the participants.

### 2.5. Data Analysis

An initial descriptive analysis was conducted by calculating the means and frequency of the variables on the total sample, the subsamples of men and women, and by age group: young adults (18–39), middle-aged adults (40–59), and older adults (over 60). For the Spanish sample, we calculated the mean ($\bar{X}$) and SD for the total sample. Participants who scored in the semi-open interval $\left[\bar{X},\;\bar{X}+SD\right)$ were considered in the mild to moderate range of distress, and those who scored above $\bar{X}+SD$ were regarded as coming within the severe range. Exploratory factor analysis (EFA) and confirmatory factor analysis (CFA) were conducted to examine the factor structure of CPDI in the Spanish sample. To this end, participants were split into two subsamples of equal size: Subsample 1 (*n* = 547) was used for EFA, and Subsample 2 (*n* = 547) for CFA, which was conducted using the lavaan package for R [35]. Findings from the hypothesized measurement model were evaluated using common indices and their cut-off points: Tucker–Lewis index (TLI) and comparative fit index (CFI) with values of ≥0.90 and ≥0.95 indicating adequate and good model fit, respectively; standardized root mean square residual (SRMR) and root mean square error of approximation (RMSEA), with values below or equal to 0.10, 0.08, and 0.05 indicating acceptable, adequate, and good model fit, respectively [36]. In addition, a cluster analysis was performed on the CPDI data used for the EFA subsample. A 2 × 3 ANOVA was performed with gender (male vs. female) and age (young, middle-aged, and older adults) as between-subjects factors using the factor scores of the CPDI as the dependent variable.

## 3. Results

The participants’ CPDI scores were as follows: 636 respondents had no distress (27.90% and 72.0% for men and women respectively), 279 participants reported mild/moderate distress (17.90% and 82.0%, for men and women, respectively), and 179 participants came in the severe distress range (7.26% men and 92.7% women). Table 2 displays mean CPDI scores as a function of gender, age group, and range of COVID-19 distress.

### 3.1. Exploratory Factor Analysis

To investigate the underlying factorial structure of the Spanish version of the CPDI questionnaire, we used Exploratory Factor Analysis (EFA). We based the analysis on a subsample (*n* = 547). All 24 items of the CPDI were subjected to a principal axis factoring and varimax rotation, Kaiser–Meyer–Olkin value was 0.91 and Bartlett’s test of sphericity was significant, Chi-square (df = 253) = 4617.65, *p* < 0.001, supporting a rationale for performing EFA. In a parallel analysis [36], an eigenvalue greater than 1 and the elbow on the scree plot was used to decide the optimal number of factors. 

The parallel analysis approach compares the eigenvalues of covariance matrices, one for the 24 data items and one based on white noise. The eigenvalues and the scree plot are shown in Figure 1. The results suggest a two-factor solution. After rotation, the first factor (Stress symptoms) accounted for 34.51% of the total variance of CPDI, while the second factor (COVID-19 Information) accounted for 14.68% of the total variance. The Stress symptoms dimension has 15 items related to negative mood, cognitive change, and physical symptoms. The COVID-19 Information factor has eight items related to attitudes to news and information in the media about the COVID-19 pandemic. Items and factor loading for the rotated factors are shown in Table 3. Factors with loadings smaller than 0.25 were omitted to improve clarity. Cronbach’s alpha for the Spanish CPDI total score was 0.88 after deleting item number 5 due to its low factor loading.

### 3.2. Confirmatory Factor Analysis

The two-factor structure obtained in the EFA was tested on the second subsample of 547 participants. We conducted CFA using the lavaan package (version 0.6-7) under R version 4 [35]. The estimation method used was Weighted Least Square (WLS). Absolute fit indices of the CFA for the Spanish CPDI were χ2df=4.59, RMSEA=0.07, GFI=0.845, AGFI=0.813, RMR=0.76 and SRMR=0.060; incremental fit indices were NFI=0.828, NNFI=0.844, and CFI=0.859, while parsimony fit indices were PGFI=0.701 and PNFI=0.749. We observed that only RMSEA, and SRMR indices were adequate and that the other indices were near the cutoff values usually considered to indicate a good fit, but they did not cross those thresholds. Using the modification indices to find a better fit was not justified theoretically because we were trying to validate the structure found in EFA with the CFA in a new sample. To this end, we modified the CFA model with the modification indices greater than 10. Although the fit indices increased, the qualitative fit results did not change. Figure 2 shows the standardized path coefficients and error variances of the original model. The standardized factor loadings ranged from 0.42 to 0.88 for the stress symptoms dimension, and from 0.47 to 0.87 for the information dimension. The correlation between the two latent variables (Stress and Information) was high (0.76).

### 3.3. Gender and Age Group Differences on Psychological Distress

We performed two 3 × 2 ANOVAs on the factor scores for Stress symptoms and COVID-19 information with Gender (men, women) and Age (Young, Middle-aged, and Older adults) as between-subjects factors. Participants were coded as young if they were aged between 18 and 39, as middle-aged if they were aged between 40 and 59, and older adults if they were aged over 60. These analyses were performed on the total sample. The results for Stress symptoms showed a main effect of Gender [(F(1, 1088)=57.029, MSe=0.856, p<0.001, $\eta_{p}^{2}=0.05$)] and a main effect of Age [(F(2, 1088)=36.182, MSe=0.856, p<0.001, $\eta_{p}^{2}=0.062$)]. As shown in Figure 3, women scored higher than men in all age groups, and older adults scored lower than young and middle-aged participants (*p* < 0.05), while there was no difference between these two groups (*p* > 0.05). 

Regarding the factor scores for Information, only the main factor of Age was statistically significant [(F(2, 1088)=28.464, MSe=0.948, p<0.001, $\eta_{p}^{2}=0.05$)]. Pair-wise comparisons showed that the three age groups differed statistically (ps>0.05). Older adults had the highest scores on COVID-19 Information, followed by middle-aged adults, and young adults had the lowest scores. The results are displayed in Figure 4.

### 3.4. Clustering

To understand the meaning of the factor scores obtained in EFA and CFA, we clustered the CPDI data used in EFA with a Gaussian mixture model (GMM) including the whole sample (*n* = 1094). This clustering technique has excellent generalizability in that each participant can uniquely and easily be mapped to a different cluster, and each point in the coordinate space is assigned a probability of being in any cluster. Mathematically, several Gaussian distributions were fitted to the density of data points in the two-dimensional factor space. We used the mclust (version 5.4.6) for R to model the data [37]. The number of clusters was determined separately using the Bayesian Information Criterion (BIC) and the Rand index using the ellipsoidal equal shape and orientation model. The results showed four clusters of participants (Figure 5) with a log-Likelihood of −2829.25, BIC equal to −5798.455, and ICL equal to −6366.736. See Figure 5.

The first factor (distress symptomatology) showed an ordered pattern of the four groups from lower to higher distress, while the second factor (COVID-19 Information) showed approximately the same mean for each cluster (around 0) but with different variance.

## 4. Discussion

In this study, we investigated the factorial structure of the CPDI via exploratory (EFA) and confirmatory factor analysis (CFA) and validated the tool in a Spanish sample. To our knowledge, the present study is the first to use a comprehensive scale making it possible to capture different aspects of psychological distress related to the COVID-19 pandemic in a Spanish sample. For this purpose, two authors of the present paper translated into Spanish and adapted the COVID-19 Peritraumatic Distress Index (CPDI). We then tested its psychometric properties conducting an exploratory factor analysis (EFA), followed by a confirmatory factor analysis (CFA) of the structure found in the EFA conducted with a different sample. The results indicated that the Spanish adaptation of CPDI has a two-factor structure, with good internal consistency and reliability. The first factor (Stress symptoms) contains items related to psychological distress, whereas the second factor (COVID-19 Information) is related to different constructs associated with negative information in social media. The only study that had been conducted to date a confirmatory factor analysis on the CPDI [32] found a one-factor solution with 19 items, after eliminating items number 5, 8, 9, 10, and 11, which did not comply with the item loading criteria (≥0.32). However, the dimension choice for the CPDI in our study, based on the elbow of the scree plot and the eigenvalues of the factors, was a two-factor solution vs. a one-factor solution. We also removed item number 5 from the analysis due to its low factor loading, while items 8, 9, 10, and 11 were maintained and loaded into the second factor above 0.30 (COVID-19 Information factor). Whereas the first factor (Stress symptoms) showed a high congruency between the Italian and Spanish samples, the second factor (COVID-19 Information) seemed to be less defined. Nonetheless, the second factor still provides valuable information about the influence of COVID-19-related news on psychological well-being. The cluster analysis performed with the factorial scores of the two CPDI factors (Stress symptoms and COVID-19 Information) revealed four clearly defined groups. These groups were defined basically as a function of the stress symptomatology, arranged from low to high degree of stress symptoms. These four clusters did not differ in the COVID-19 Information factor. The results also showed that the higher the stress symptomatology, the more homogeneous were the groups.

Concerning the prevalence and severity of COVID-related psychological distress in the Spanish population and based on the cut-off values of distress in CPDI, in our Spanish sample, 279 participants (25.5%) reported mild to moderate distress and 179 participants (16.36%) experienced severe distress, compared to 52.0% and 18.8% in Brazil [26], 47.0% and 14.1% in Iran [22], and 29.3% and 5.1% in China [9]. The average CPDI score of our Spanish-based sample (Mean = 26.15, SD = 15.28) was higher than that reported in China (Mean = 23.65, SD = 15.45), Italia (Mean = 18.61, SD = 12.2), and Germany (Mean = 21.9, SD = 12.6), but lower than that reported in Brazil (Mean = 37.6, SD = 15.22) [26], Iran (Mean = 34.54, SD = 14.92) [22], and France (24.01, SD = 12.76) [28]. Furthermore, we found gender differences in the mild/moderate distress range (17.92% and 82.07% for men and women, respectively) and the severe distress range (7.26% and 92.73% for men and women, respectively). These findings are in line with those obtained by other studies using the CPDI [9,23,24,25,26,27]. Furthermore, studies using instruments other than the CPDI to measure stress symptoms reported similar findings, with women experiencing higher levels of psychological distress than men [8,13,15,38]. It might be possible that females are more predisposed to express their emotional states. However, as these findings were reported across different cultures, more research is needed to understand what causes the gender differences.

In this study, we additionally analyzed the gender differences using the factorial scores of the CFA. The results revealed that the gender differences were only related to the Stress symptoms factor and not to the COVID-19 Information factor, independently of the age group. Women experienced more physical symptoms, negative mood, and behavioral changes than men, irrespectively of age. The absence of gender differences in the Information factor suggests that both, men and woman, are similarly reactive towards COVID-related information, but that this information might affect women more negatively. Our results indicate that with increasing age, people tend to be more susceptible for COVID-related information input. Paradoxically, this age-related increase in Information is not associated with an increase in stress symptoms, which are lower in older than in younger adults.

Age is an important risk factor for mortality due to COVID-19, increasing exponentially from 50 years on [39]. Although people over 60 should be more concerned about getting infected, because they are considered at high risk, in our sample participants over 60 showed lower levels of distress than young and middle-aged participants, with no difference between the latter groups. This result could be explained by the fact that older adults are less affected than other age groups by the economic impact of the pandemic (i.e., unemployment, reduced turnover, etc.), and carry less burden of family-related responsibilities. Furthermore, it seems that older people find it easier to relativize the traumatic impact due to their life-long experience. Our results on age differences are in line with those found by other studies. For example, García-Fernández et al. [40] found in a Spanish sample, that older adults were less vulnerable than younger participants to suffer from depression and acute stress, whereas no age differences were found in anxiety during the peak of the pandemic. Age-related differences were also found in a Canadian sample [41], with significantly lower mean scores for stress, anxiety, and depression in older than in younger adults.

Another study [42] examined the association between age, COVID-19 disruption, stress, and affect. In this study, middle-aged and older adults experienced less distress than younger adults in response to their perceived life disruptions caused by the pandemic. The authors suggest, that when faced with a stressor (global pandemic), middle-aged and older adults might regulate better their emotions, even when they perceive the stressor as disruptive. Therefore, older adults seem to adapt better to the crisis caused by the pandemic. In this sense, López et al. [43] investigated the psychological well-being (personal growth and purpose in life) experienced during the COVID-19 crisis in a Spanish sample of young–old (60–70 years) and old–old adults (71–80) and the variables associated. Although differences were found between both older age groups (young–old experienced more personal growth, but they did not experience more purpose in life than old–old), the results of this study showed that personal resources, such as resilience and gratitude, and less experiential avoidance, are related with greater levels of personal growth.

Regarding the Spanish adaptation of the CPDI, more research is needed to understand the discrepancy between the factorial solutions found in the Spanish and Italian samples. Although this discrepancy could be due to differences between samples, we think that they were due to intrinsic features of the questionnaire. In the present study, we found that even statistical indices that assessed the number of dimensions of the questionnaire did not agree. For example, the parallel analysis showed nine dimensions, while the scree plot showed two clear dimensions. We chose a two-factor solution as a compromise between the results of the scree plot, the eigenvalue value of 1 and the parallel analysis. The moderate fit indices in the CFA in our study also suggest that the factor solution for CPDI is inconclusive and that the items of the CPDI need to be reviewed in the European population.

## 5. Limitations

The present study is not without limitations. As data was only collected at one point in time, the results do not permit causal inferences to be drawn. To evaluate the impact of the pandemic outbreak on psychological well-being in terms of a change produced in the person, would need longitudinal data, especially from before the pandemic outbreak. Moreover, the procedure of data collection inherently implies sampling biases, as only those people who disposed of the necessary technical devices to complete the survey could participate. This could explain why our sample was biased towards a higher academic background. In our study, approximately half of the data came from the Autonomous Community of Madrid. This area was the most affected during the first wave of the pandemic, which could explain in part the greater participation in Madrid. However, the snowball-sampling method might accentuate biases, not only towards a geographical area, but also towards certain age groups or socio-demographic backgrounds. Moreover, the fit indices for the CFA model were just below the threshold value considered acceptable. The just below fit shows that there is potential for improvement in the CPDI scale in future studies.

## 6. Conclusions

Women experience more COVID-related distress than men, and the level of distress decreases with age. The gender differences in the CPDI scores are related with items associated with stress symptoms, but not with those associated with COVID-19 related information input. This suggest that, even though both genders are exposed to similar information about COVID-19, women might be more prone to process this information negatively. The CPDI constitutes a promising tool for capturing the psychological impact of the current health crisis and might be useful for a rapid detection of potential peritraumatic stress caused by the COVID-19 pandemic. Future studies should include the development of more comprehensive measures adding items about risk, vulnerability, and protective factors that contribute to psychological distress.

## Figures and Tables

**Figure 1 ijerph-18-05253-f001:**
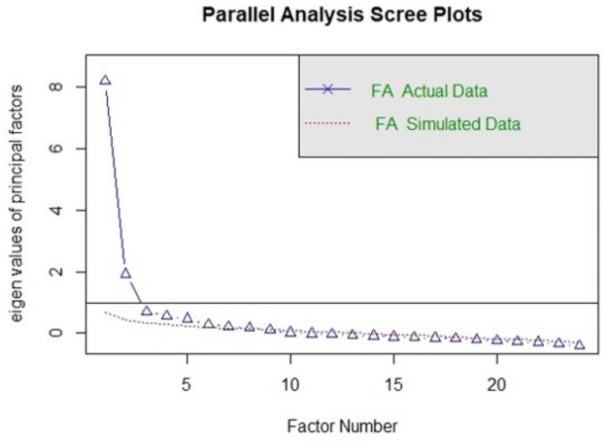
Parallel analysis scree plots.

**Figure 2 ijerph-18-05253-f002:**
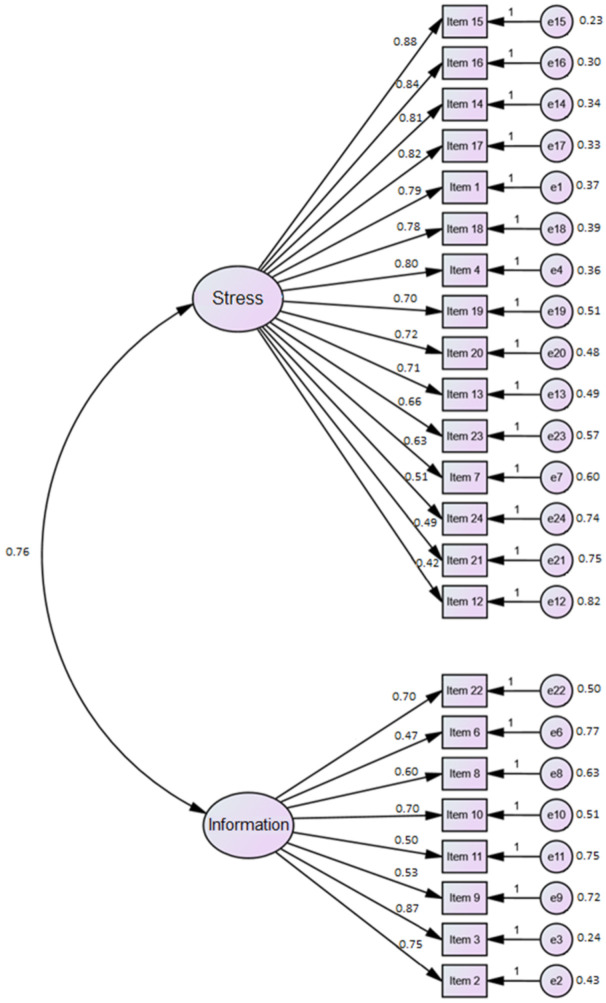
The standardized factor loadings and error variances for the hypothesized model.

**Figure 3 ijerph-18-05253-f003:**
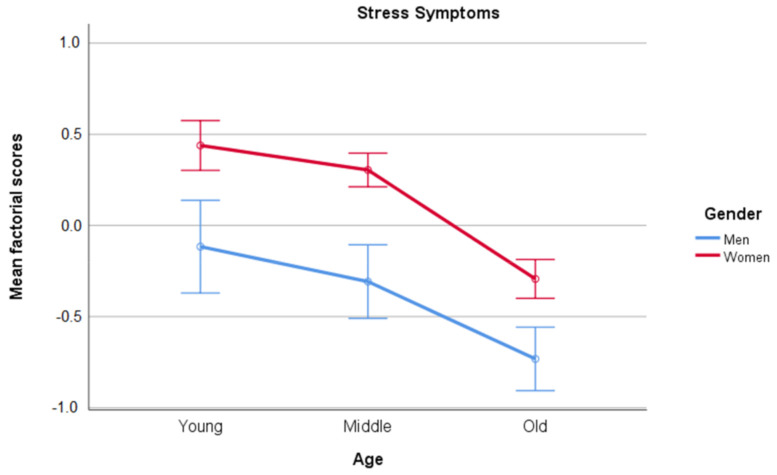
Mean scores and confidence interval (95%) for factor scores in Stress symptoms as a function of participants’ Gender and Age.

**Figure 4 ijerph-18-05253-f004:**
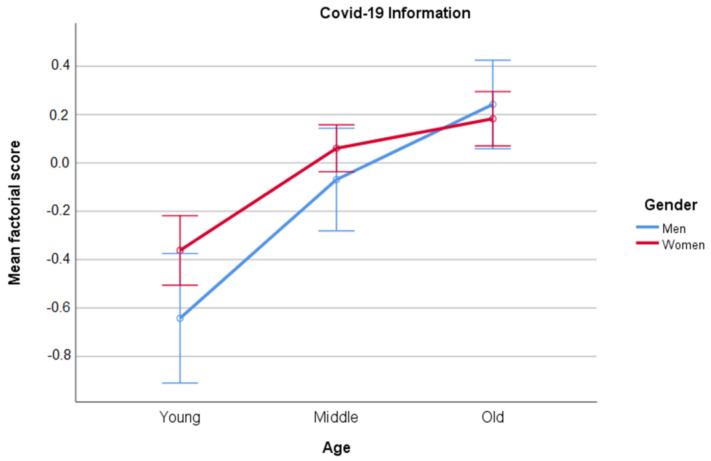
Means and confidence interval (95%) of factor scores for COVID-19 Information as a function of Gender and Age.

**Figure 5 ijerph-18-05253-f005:**
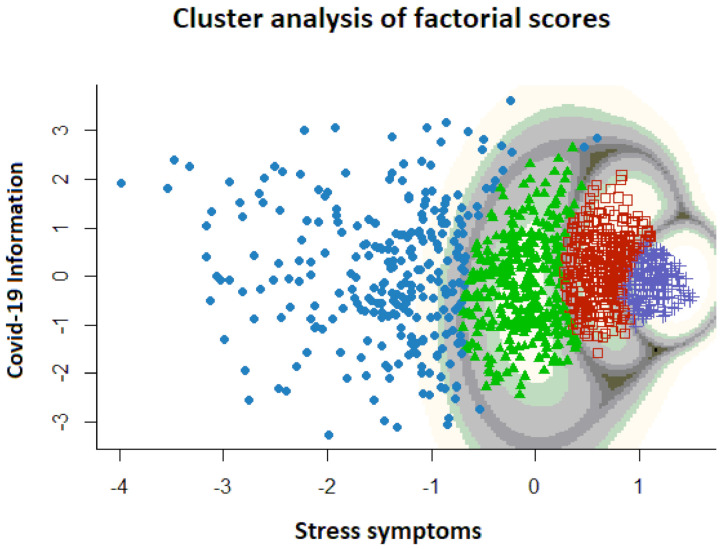
Results of the cluster analysis of the CPDI data conducted with the whole sample.

**Table 1 ijerph-18-05253-t001:** Description of socio-demographic background variables. Percentages and SDs are shown in parentheses.

Variable	Level	*n* (%)	M (SD)
Gender	Men	241 (22%)	
	Women	853 (78%)	
Age	Total		52.55 (14.19)
	18 to 39 years	228 (20.8%)	31.69 (5.87)
	40 to 59 years	468 (42.8%)	49.82 (5.88)
	≥60 years	398 (36.4%)	67.04 (5.22)
Education ^1^	Total		5.2 (1.18)
	University degree or higher	658 (60.1%)	
	High school diploma	181 (16.5%)	
	Vocational training	134 (12.2%)	
	Senior high school	62 (5.7%)	
	Junior high school or less	59 (59%)	
Employment status ^2^	Total		1.46 (0.81)
	Employed	747 (68.3%)	
	Unemployed or housekeeper	66 (6%)	
	Student	22 (2%)	
	Retired	259 (23.7%)	
Place of residence	Autonomous Community of Madrid	538 (49.2%)	
	Rest of Autonomous Communities	379 (34.6%)	
	N/A	177 (16.2)	
Professional occupation ^3, 4^	Armed forces occupations	6 (0.8%)	
	Clerical support workers	169 (22.6)	
	Craft and related trades workers	7 (0.9%)	
	Elementary occupations	32 (4.3%)	
	Managers	13 (1.7%)	
	Professionals	317 (42.4%)	
	Service and sales workers	76 (10.1%)	
	Technicians	87 (11.6%)	

^1^ Level of educational attainment was defined as follows: 1 = below primary education, 2 = junior high school, 3 = senior high school, 4 = vocational training, 5 = high school diploma, 6 = university degree or higher. ^2^ Employment status was defined as follows: 1 = employed, 2 = retired, 3 = student, 4 = unemployed or housekeeper. ^3^ Only active workers were included in this section (*n* = 747). ^4^ Professional occupations were categorized according to the 10 mayor groups of the International Standard Classification of Occupations (ISCO).

**Table 2 ijerph-18-05253-t002:** Mean values of CPDI scores as a function of gender, age group, and range of COVID-19 distress. SDs are shown in parentheses.

	*n* (%)	M (SD)
Total Sample	1094	26.15 (15.28)
Gender		
Men	241 (22%)	19.75 (12.37)
Women	853 (77%)	27.95 (15.53)
Age groups		
Young Adults	228 (20.8)	26.46 (15.69)
Middle-aged Adults	468 (42.7)	29.01 (16.23)
Older Adults	398 (36.3)	22.61 (13.06)
Range of COVID-19 distress		
No Distress	636 (58.13)	15.59 (5.93)
Mild to moderate	279 (25.50)	33.11 (4.29)
Severe	179 (16.36)	53.84 (9.50)

**Table 3 ijerph-18-05253-t003:** Factor loadings for the rotated factor.

Items	Factor Loading		Communality
	1	2	
15. Due to feelings of anxiety, my reactions are becoming sluggish	0.811		0.681
16. I find it hard to concentrate	0.772		0.602
14. I feel tired and sometimes even exhausted	0.770		0.593
17. I find it hard to make any decisions	0.727		0.568
1. Compared to usual, I feel more nervous and anxious	0.703		0.542
18. During this COVID-19 period, I often feel dizzy or have back pain and chest distress	0.715		0.519
13. I am more irritable and have frequent conflicts with my family	0.701		0.495
4. I feel empty and helpless no matter what I do	0.657	0.308	0.527
19. During this COVID-19 period, I often feel stomach pain, bloating, and other stomach discomfort	0.612		0.380
20. I feel uncomfortable when communicating with others	0.634		0.416
23. I lost my appetite	0.590		0.363
7. I am losing faith in the people around me	0.437		0.241
24. I have constipation or frequent urination	0.422		0.178
21. Recently, I rarely talk to my family	0.366		0.138
22. I cannot sleep well. I always dream about myself or my family being infected by COVID-19	0.350	0.385	0.271
12. I avoid watching COVID-19 news, since I am scared to do so	0.332		0.113
6. I feel helpless and angry about people around me, governors, and media	0.331	0.338	0.244
8. I collect information about COVID-19 all day. Even if it’s not necessary, I can’t stop myself		0.720	0.518
11. I am constantly sharing news about COVID-19 (mostly negative news)		0.654	0.439
3. I can’t stop myself from imagining myself or my family being infected and feel terrified and anxious about it	0.315	0.632	0.498
2. I feel insecure and bought a lot of masks, medications, sanitizer, gloves and/or other home supplies		0.614	0.413
9. I will believe the COVID-19 information from all sources without any evaluation		0.612	0.375
10. I would rather believe in negative news about COVID-19 and be skeptical about the good news		0.552	0.390

Note. Items with loading < 0.25 were omitted.

## Data Availability

Materials and data for this study are available on the project page located on the Open Science Framework (https://osf.io/jkw6b/, accessed on 13 May 2021).

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
