# Peer review of "COVID-19 Peritraumatic Distress as a Function of Age and Gender in a Spanish Sample"

_ijerph, 2021, doi:10.3390/ijerph18105253_

Round 1
Reviewer 1 Report
The authors have adequately addressed the comments and concerns I raised in my initial review. The manuscript has been significantly improved and now warrants publication in International Journal of Environmental Research and Public Health. However in my previous review I suggested
a.- The goodness of fit index (GFI) and the adjusted goodness of fit index (AGFI) with a value of over 0.9 generally indicating acceptable model fit. These values in the present study are ??? =0.845 and ???? = 0.813. Could you include this in the limitations of your study?
b.- Discussion: The authors are advised to discuss in depth about the age differences. Some previous Spanish studies found a good emotional response among older adults to the Covid-19 (e.g. López et al., 2020; Losada-Baltar et al., 2020). Approximately half of the data in the present study came from the Autonomous Community of Madrid. Nevertheless authors included in the reviewed paper a Portuguese study. Could you include any Spanish studies?
c.- Is the sample representative of Spanish population? Could you discuss this briefly?
d.- Why the authors did not conduct test-retest reliability? Could you discuss this briefly?
e.- Is the study supported by a grant from the Spanish Ministry of Economy and Competitiveness (grant # PSI2016-80377-R) to SB and JMR (Effects of multidomain versus unidomain training in executive control and memory of older adults)?
Is the study supported by a grant from the Council of Madrid (B2017/BMD-3688) to JMR (IMAGEN MULTIMODAL DE LA RESPUESTA TERAPÉUTICA A ESTRATEGIAS MULTIDIANA EN ENFERMEDADES NEUROLÓGICAS)?
Recommendation: Despite this, I thing that from a qualitative viewpoint, the manuscript is ready for publication. I congratulate the authors on this useful contribution to the literature.
Author Response
We thank Reviewer 1 for the encouraging comments. Please, see our response to your comments.
However in my previous review I suggested
a.- The goodness of fit index (GFI) and the adjusted goodness of fit index (AGFI) with a value of over 0.9 generally indicating acceptable model fit. These values in the present study are ??? =0.845 and ???? = 0.813. Could you include this in the limitations of your study?
RESPONSE
Thank you for noticing that. We added a paragraph on this issue (see the Limitation section of the reviewed manuscript).
b.- Discussion: The authors are advised to discuss in depth about the age differences. Some previous Spanish studies found a good emotional response among older adults to the Covid-19 (e.g. López et al., 2020; Losada-Baltar et al., 2020). Approximately half of the data in the present study came from the Autonomous Community of Madrid. Nevertheless authors included in the reviewed paper a Portuguese study. Could you include any Spanish studies?
RESPONSE
Thank you for the comment. In response to your suggestion, we broadened the Discussion section, discussing in-depth the issue of age differences. The novelty of our article resides in the analysis of the factorial structure of the CPDI. We also discussed the age differences as a function of the found two-factor structure. After a comprehensive rereading, we decided to eliminate the Portuguese study from our article, as we understood that it did not fit with the objectives of our work, and incorporated the work of López et al., 2020 which you kindly suggested.
c.- Is the sample representative of the Spanish population? Could you discuss this briefly?.
RESPONSE
We cannot assure that the sample is statistically representative for the Spanish population. However, we consider that, for our study, the different age groups are adequately represented in our sample.
d.- Why the authors did not conduct test-retest reliability? Could you discuss this briefly?
RESPONSE
Thank you very much for your comment. As it is known, there exist several indices of reliability and within these, we opted for Cronbach’s alfa (p. 8, lines 270-272). The calculation of the test-retest reliability would only be possible within a longitudinal study design, which was not the objective of the present work. Notwithstanding, we know that. So, we can be sure that our test-retest reliability coefficient would be greater or equal to 0.88. This value ensures a high reliability for the test.
e.- Is the study supported by a grant from the Spanish Ministry of Economy and Competitiveness (grant # PSI2016-80377-R) to SB and JMR (Effects of multidomain versus unidomain training in executive control and memory of older adults)?
RESPONSE
Yes, this work is supported by the mentioned grant.
Is the study supported by a grant from the Council of Madrid (B2017/BMD-3688) to JMR (IMAGEN MULTIMODAL DE LA RESPUESTA TERAPÉUTICA A ESTRATEGIAS MULTIDIANA EN ENFERMEDADES NEUROLÓGICAS)?
RESPONSE
Yes, this work is supported by the above-mentioned grant.
Recommendation: Despite this, I thing that from a qualitative viewpoint, the manuscript is ready for publication. I congratulate the authors on this useful contribution to the literature.
RESPONSE
Thank you very much for this comment.
Reviewer 2 Report
There are typing and grammar issues in the abstract. The abstract must be clear, concise and standing alone. I am not sure about the outputs and the scientific impact having the gender as the center of distress.
In the introduction is needed a stronger support for the hypothesis. The correlations presented are clear, but the need for studying it and to place it in a research journal paper is not well presented, stated and supported.
Limitations of the study are now mentioned. But there is no proper scientific evidence of the results of the manuscript. I cannot find scientific relevance in the manuscript. There is not also stated in it.
There is no a clear explanation of how this research would help public health authorities to implement effective approaches.
Author Response
There are typing and grammar issues in the abstract. The abstract must be clear, concise and standing alone. I am not sure about the outputs and the scientific impact having the gender as the center of distress.
RESPONSE
Thank you very much for your comment. The manuscript was proofread twice by two native English speakers and we are confident that all grammar and typing issues were resolved. Two factors that have been repeatedly associated with differential distress scores are gender and age, justifying their inclusion into the analysis. In the present study, we showed that men and women not only produce different CPDI scores but also that gender differences only apply to one of the two factors we found in the CPDI. This result opens the door for further research, targeting more precise analyses of constructs that might affect men and women in different ways.
In the introduction is needed a stronger support for the hypothesis. The correlations presented are clear, but the need for studying it and to place it in a research journal paper is not well presented, stated and supported.
RESPONSE
Thank you very much for this constructive comment. We agree with you that the interpretation of the results was insufficient and that the objectives of this study were not clearly defined. Therefore, we revised the entire manuscript and made different modifications and updates in the Introduction, Discussion, and Conclusion sections. Our intention was to provide a more coherent information flow and to make our goals, findings, and their interpretation more understandable. We hope that these improvements meet your expectations.
Limitations of the study are now mentioned. But there is no proper scientific evidence of the results of the manuscript. I cannot find scientific relevance in the manuscript. There is not also stated in it..
RESPONSE
As we mentioned in the previous point, the entire document, especially the interpretation of the results, was subjected to an exhaustive revision. We hope that the objectives, findings, and their interpretation are now well-defined and that the contribution of our work to the scientific knowledge pool is clearer now.
There is no a clear explanation of how this research would help public health authorities to implement effective approaches
RESPONSE
Our study belongs to basic research, i.e., we produce knowledge which in the next step might be incorporated into applied research designs. Thus, it is beyond the scope of the present study to provide recommendations on efficient health care approaches.
Reviewer 3 Report
I thank the authors for taking time to revise their manuscript considering my concerns.
All points I raised were managed or are now described as limitations of the paper.
Author Response
I thank the authors for taking time to revise their manuscript considering my concerns. All points I raised were managed or are now described as limitations of the paper.
RESPONSE
Thank you very much for this encouraging comment.
This manuscript is a resubmission of an earlier submission. The following is a list of the peer review reports and author responses from that submission.
Round 1
Reviewer 1 Report
This study appears to be a well-executed psychometric evaluation of the Covid-19 Peritraumatic distress in a Spanish sample. However, it is not clear how this paper differs from previous studies which have also evaluated the Covid-19 impact in Spain. This manuscript is well-written. There are some comments and questions for authors to answer and revise the manuscript if possible.
Major
Methods: Please include the response rate of the survey.
The original questionnaire is written in English. Are there back translations and reviews? Could authors of this study be confident of its linguistic validity? University graduates constitute 60.1% of the sample and this paper only offers a global description.
The goodness of fit index (GFI) and the adjusted goodness of fit index (AGFI) with a value of over 0.9 generally indicating acceptable model fit. The values in the present study are ??? =0.845 and ???? = 0.813.
Minor
The source of subjects is not clearly described. How did the authors select subjects from each pool of population? Is the sample representative? We may expect more details of demographic characteristics.
Could you, please, be more specific about the study purpose in the Introduction?
Why did you choose to recruit a convenience sample? Please, justify.
How does your sampling approach bias the results?
I am not sure the background laid a solid foundation for expecting these outcomes to differ by gender/sex
Tables may have a footer. Please include “SDs are shown in parentheses” in Table 1 footer.
Statistical analyses and tests:
(1) The authors did not conduct test-retest reliability.
(2) Clinical validity or known groups comparison: There were no anchors of clinical condition (known groups) or change over time.
Discussion: The authors are advised to discuss in depth about the age differences. Some previous Spanish studies found a suitable emotional response among older adults to the Covid-19:
- Losada-Baltar A, Jiménez-Gonzalo L, Gallego-Alberto L, Pedroso-Chaparro MDS, Fernandes-Pires J, Márquez-González M. “We're staying at home”. Association of self-perceptions of aging, personal and family resources and loneliness with psychological distress during the lock-down period of COVID-19. J Gerontol B Psychol Sci Soc Sci. (2020) gbaa048. 10.1093/geronb/gbaa048
- López J, Perez-Rojo G, Noriega C, Carretero I, Velasco C, Martinez-Huertas JA, et al. . Psychological well-being among older adults during the COVID-19 outbreak: a comparative study of the young–old and the old–old adults. Int. Psychogeriatr. (2020) 1–6. 10.1017/S1041610220000964
Why did they find two domains? The contribution is not clearly stated.
Please include some limitations of your study (e.g. the early stage of the pandemic when the data were collected, the sampling procedure, the exploratory nature of the data collected).
Is the study supported by a grant from the Spanish Ministry of Economy and Competitiveness (grant # PSI2016-80377-R) to SB and JMR (Effects of multidomain versus unidomain training in executive control and memory of older adults)?
Is the study supported by a grant from the Council of Madrid (B2017/BMD-3688) to JMR (IMAGEN MULTIMODAL DE LA RESPUESTA TERAPÉUTICA A ESTRATEGIAS MULTIDIANA EN ENFERMEDADES NEUROLÓGICAS)?
Please review the references. Some references have no number:
- Mazza, C.; Ricci, E.; Biondi, S.; Colasanti, M.; Ferracuti, S.; Napoli, C.; Roma, P. et al. (2020). A Nationwide Survey of Psychological 450 Distress among Italian People during the COVID-19 Pandemic: Immediate Psychological Responses and Associated Factor. 451 J. Environ Res. Public Health, 2020,17, 3165. doi:10.3390/ijerph.17093165 452
- Bonanad, C;, García-Blas, S.; Tarazona-Santabalbina, F.; Sanchis, J.; Bertomeu-González, V.; Fácila, L.; Ariza, A.; Núñez, J.; Cordero, 453 The effect of age on mortality in Patiens with COVID-19. A Meta-Analysis with 611.583 subjects. J Am Med Dir Assoc. 2020, 454 21, 7, 915-918. 455
- Qualtrics and all other Qualtrics product or service names are registered trademarks or trademarks of Qualtrics, Provo, 456 UT, USA. Copyright © [2020]; https://www.qualtrics.com
Some references have a wrong number (e.g. reference number 33 has been included in the text before the 24-30 references)
Reviewer 2 Report
The present study provides a study-related on Covid-19 Peritraumatic distress as a function of age and gender in a Spanish sample
In this submitted preliminary version, this reviewer can see that there are some typing and grammar issues In the manuscript. I suggest making it review again, with a native English speaker.
I consider this work exciting but not relevant as an Article. Maybe for Communication. But the study lacks novelty in terms of methods/sample size (1,094) and the way they land into the discussions. I can not find the scientific soundness of this work.
General comments:
The full introduction and results sections need a full and more in-depth review. It is hard to read. And most important, the objectives are not very well defined. There is no clear explanation of how these results/conclusions would help public health in the results and discussions. I could not find the limitations of the study. This is really important to be defined.
Discussion needs to be improved. There is a lack of connecting points in results and evidence.
Furthermore, the presented tool can help to define factors that contribute to psychological distress. CPDI is a suitable screening tool for the rapid detection of potential peritraumatic stress caused by the Covid-19 pandemic.
Reviewer 3 Report
The paper deals with an interesting topic and is globally well written.
The paper describes results obtained using CPDI in Spain, administering the test to a relatively large group (1094 subjects). A confirmational factorial analysis is provided, as well as a discussion of the role of sex.
Strenghts are that the analysis is well conducted and that the number of subjects is adequate; the paper is globally well written.
I have however a few concerns to raise.
First: no mention of approval by an Ethic Committee is given.
Second: was a priori sample size calculated? How? Or why not?
Third: no limitations are mentioned. However the authors used only one questionnaire without direct clinical examination and used a way to obtain the sample that could have caused biases (since it is based on voluntary choice); all this in the context of a cross-sectional observation.
Fourth: the claim that the CPDI "adequately captures the psychological impact of the current health crisis experienced by Spanish men and women" in not sufficiently supported by available data, considering limitations of the study.